# UPDATE LARGER, TRAIN FASTER: STABLE TEST-TIME ADAPTATION UTILIZING NOISY-PSEUDO LABELS

## ABSTRACT

We investigate the role of pseudo-labels in the test-time adaptation (TTA) problem. When working with unlabeled samples in TTA, pseudo-labels have become a natural approach to updating the target model. However, pseudo-label learning also presents some challenges: it suffers from a memorization effect (the model learns from clean labels first, then memorizes the noisy ones) and confirmation bias (errors from noisy labels increase over time and disrupt model performance when they become significant). Our work first identifies two underlying mechanisms leading to these obstacles. On the one hand, existing methods follow a "slow" adaptation to the target domain, allowing sufficient time for the model to memorize noisy labels (memorization effect) and accumulate errors (confirmation bias). Furthermore, training with noisy labels blurs the decision boundary with nearby classes. To address the first issue, we propose a novel loss function, namely sparse cross logit (sparse-CL), that operates in the logit space and allows the model to take larger learning steps in a stable training manner. This helps the target model reach a better solution faster under the same number of updating steps. To address the second issue, we introduce a regularization that penalizes negative pseudo-labels while encouraging positive ones, which can increase the boundary between nearby classes. We demonstrate that our methods outperform state-of-the-art methods in a diverse set of TTA experiments.

## 1 INTRODUCTION

Distribution shift – when the data distribution during inference differs from the training data, is a common real-world scenario. Under distribution shift, it has been shown that a pre-trained model can degrade its performance significantly (Quinonero-Candela et al., 2008; Koh et al., 2021; Fang et al., 2020). Domain adaptation is one of the most dominant settings for distribution shift, wherein the learner aims to perform well in a target domain given access to both labeled training data from source domains and (unlabeled) data from the target domain (Ben-David et al., 2010). However, the learner cannot access the pre-training data during inference in many privacy-sensitive applications, rendering domain adaptation approaches obsolete. Test-time adaption (TTA) is a more practical setting of distribution shift wherein the learner aims to adapt to any new unlabeled input data from a target domain to make a prediction using only a pre-trained model, without access to the training data during inference.

Due to the wide range of practical demands, many different methods have been proposed and have achieved impressive results in TTA. TENT (Wang et al., 2020), inspired by the assumption that the batch normalization (BN) layer is responsible for scaling and shifting neural network features (Li et al., 2016), merely trains the BN layers of the target model based on minimizing self-entropy. This approach is then extended to EATA (Niu et al., 2022) and SAR (Niu et al., 2023), where SAR addresses limitations in TENT by introducing new techniques called reliable sample selection and sharpness-aware loss to adapt to real-world scenarios. Additionally, works by (Zhao et al., 2023; Lim et al., 2023) indicate that the current estimation of BN is inaccurate, and they suggest a new one. On the other hand, the lack of labels in the target data has encouraged the use of self-learning (Gandelsman et al., 2022; Jang & Chung, 2022), pseudo-label learning (Goyal et al., 2022; Liang et al., 2020), and self-supervised learning (Chen et al., 2022; Zhang et al., 2022b) as effective approaches. Among these approaches, pseudo-labeling is the most preferable because it naturally follows the training paradigm (training on supervised cross-entropy with true labels). Additionally,

Figure 1: Our main architecture works as follows: For each input test image, we extract its logit vector through the target model. This vector is then sorted in descending order of logit magnitude. Based on this sorted vector, the class with the highest logit value (index 1) is selected as the pseudo-label and trained using the sparse cross logit function (sparse-CL). Next, we skip the class indices from 2 to $(s + 2)$ to reduce the noisy effect of negative loss. Finally, classes from indices $(s + 3)$ to $(s + k + 3)$ are selected as complementary labels for the k-hardness negative learning function (k-NL).

the cost of generating pseudo-labels is low, making it more computationally efficient. Finally, in a pseudo-learning setting, techniques like label smoothing (Liang et al., 2020) and label refinement (Chen et al., 2022; Wang et al., 2022) have shown promising results in improving label quality, therefore enhancing model adaptation performance.

Despite recent achievements in TTA utilizing pseudo-labels, working under this setting still presents four obstacles:

- Memorization effect (Arpit et al., 2017): During the pseudo-label generation step (where the target model predicts sample labels), some labels will inevitably be noisy. During tuning, the target model learns from clean samples first, then memorizes the noisy labels, which deteriorates model performance (Yi et al., 2023; Arpit et al., 2017; Liu et al., 2020).
- Confirmation bias (Arazo et al., 2020): Errors in noisy labels accumulate over time, leading to the failure of the learned model.
- Unbounded noise ratio: As indicated in (Liu et al., 2020), the noise in pseudo-labels is unbounded, whereas recent methods that work on robust noise are constrained by a bounded noise ratio (Song et al., 2022; Natarajan et al., 2013). Therefore, there is a lack of theoretical guarantees for applying common robust noise methods to pseudo-label learning.
- Adaptation cost: TTA is expected to work well in an online manner, which means we need to trade off between the computation cost and the accuracy of the target model when adapting to new samples. An acceptable model should achieve high performance with just one or a few adaptation steps.

Our work hypothesizes that these challenges can be solved efficiently if the adaptation process is conducted in a "quick" learning manner. Quick learning here means that, under the same number of training steps (including forward and backward steps), the algorithm will guide a model to converge faster to an optimal point compared with the original training. This strategy accelerates adaptation, enabling the model to partially avoid memorizing noisy samples through the early learning phenomenon (Liu et al., 2020). Furthermore, larger learning steps facilitate rapid convergence to the target domain, reducing the adaptation gap and cost. From this perspective, the generated pseudo-label procedure becomes more accurate with a lower noise ratio compared to standard training, as labels are produced at each inference step, benefiting from improved convergence speed. Lastly, updating the model with larger steps requires fewer iterations to achieve a satisfactory target model, mitigating the impact of confirmation bias, where errors accumulate proportionally to the number of updating iterations.

Naturally, we can achieve this "quick" adaptation goal by increasing the magnitude of the learning rate. However, a larger learning rate can make the network unstable during training. For example,

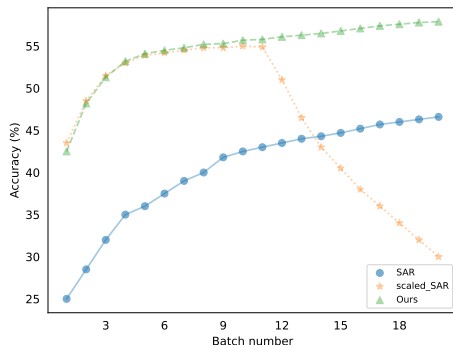

Figure 2: The learning curve (measured by test accuracy on Gaussian noise corruption level 5, ImageNet-C) of SAR under normal learning rate (blue), high learning rate (green), and our method (orange). Generally, our method can learn stably under a large learning rate and yield better performance. Similarly, entropy minimization (SAR) improves gradually under a small learning rate during adaptation. However, when we scale up the learning rate in SAR (scaled-SAR) to match the one used in our loss, the target model updates faster in the first few steps, but then it suddenly falls and converges to a poor solution.

in Figure 2, we show that entropy minimization can achieve higher results with a large learning rate in the initial steps, but it crashes afterward due to large updates. Therefore, to learn with a large learning rate, we need a training strategy that is stable enough to tackle the instability problems associated with large updates. From this perspective, previous works have shown that loss surface and gradient variance are key factors that control model stability (Hardt et al., 2016; Zheng et al., 2016; Foret et al., 2020; Li et al., 2018; Agarwal et al., 2022). On the one hand, saddle loss landscapes have been analyzed both theoretically and empirically, proving that they can improve model generalization (Foret et al., 2020; Hardt et al., 2016; Kwon et al., 2021; Liu et al., 2022). Additionally, small gradient variance plays an important role in the stability of the learning process (Johnson & Zhang, 2013; Agarwal et al., 2022; Liu et al., 2021), where small gradient variance means that at each updating step, the gradient does not vary too much, making the learning process more stable. In this work, we follow the second approach to define a stable learning strategy by introducing a new loss function named sparse-CL, which operates in the logit (pre-softmax) space and can yield stable gradient norms between batches (or samples) during adaptation. We then investigate a negative learning method called $k$-NL, which can be jointly trained with sparse-CL to improve model performance by enlarging the distance between each sample and its negative classes. (Figure 1).

Our works are organized as follows:

- We investigate factors that can affect the stability during adaptation, then introduce the sparse cross-logit learning method named sparse-CL, which works on logit space and helps stabilize the learning process.
- Then, we devise a new negative learning method named $k$-NL that can further boost the performance of the sparse-CL loss.
- Finally, we empirically show that our method can work stable with a large learning rate, leading to an efficient solution with memorization effect and accumulation error, and enhance the target model to achieve new SOTA results with a large improvement (improving over 6% on average in various settings compared to SAR).

## 2 BACKGROUND

**Test-time adaptation** Given the source data $\mathcal{X}_s = \{x_1, x_2, \cdots, x_n\}$ following the source distribution $P(x)$ and the labels $\mathcal{Y}s = \{y_1, \ldots, y_n\}$, which have $C$ classes where $y_i \in \{1, 2, \cdots, C\}$, we acquire a trained source model $\mathcal{F}s(\cdot)$ adapting to the source data during the source training phase. After that, we utilize knowledge from the source model to transfer to a new target domain $Q(x)$

to achieve a target model $\mathcal{F}_t(\cdot)$, where unlabeled test samples $\mathcal{X}_t = \{x_1, x_2, \cdots, x_m\}$ follow the distribution $Q(x)$, and a target label $\mathcal{Y}_t = \{y_1, \ldots, y_m\}$, which needs to be inferred during the adaptation. In this setting, $Q(x) = t(P(x))$, where $t(\cdot)$ is a transformation that shifts a sample from the source domain $P(x)$ to the target domain $Q(x)$. Under the TTA setting, we split the target data into $\mathcal{T}$ batches, where each batch includes $\mathcal{B}$ samples. The target model is initialized from the source model and will train on each batch of data in $\mathcal{S}$ steps before making predictions. The model will continue until all of the data have been trained on and inferred. Moreover, the learned information from previous batches will be used to help predict the next batch (continual learning). Additionally, we focus on online TTA, where the target model observes each sample only once, which means $\mathcal{S} = 1$.

**Setting**  Our work decomposes the source model $\mathcal{F}_s(\cdot)$ into two components: the feature extractor $f_s(\cdot)$ and the classifier $h_s(\cdot)$, which is used to make a final prediction. The target model $\mathcal{F}_t(\cdot)$ is initialized from the source model $\mathcal{F}_s(\cdot)$, which means $f_t(\cdot) = f_s(\cdot)$ and $h_t(\cdot) = h_s(\cdot)$. For each input sample $x$, we extract its representation vector $f_x = f(x)$. This vector is then input into the classifier $h(\cdot)$ to obtain the logit vector $h_x = h(f_x)$. The pseudo label of the sample $x$ is then constructed from this logit vector by applying the one-hot operation on the softmax output: $\hat{y} = O(p_x)$, where $p_x = S(h_x)$ is the softmax vector, and $O(\cdot)$ and $S(\cdot)$ are the one-hot and softmax functions, respectively. The pseudo label $\hat{y}$ for each sample $x$ is the one-hot vector created from the output of the classifier $h_s(\cdot)$. Additionally, we view the classifier $h(\cdot)$ as a list of $C$ prototypes $P = \{f_1, f_2, \cdots, f_C\}$, where $f_i, i \in \{1, 2, \cdots, C\}$, is the prototype of class $i$.

## 3 METHOD

### 3.1 GRADIENT ANALYSIS

In this section, we will analyze the gradient of two common loss functions used in TTA: entropy minimization and cross-entropy, to understand their gradient behavior during the training process. For each input sample $x$, we extract its logits vector as $h = h_t(f_t(x))$. Then, the softmax vector and pseudo labels are defined as $p = S(h)$ and $\hat{y} = O(p)$, respectively.

**Entropy minimization**  Based on the above setting, the entropy minimization loss for sample $x$ can be defined as:

$$\mathcal{L}_{EM} = \sum_{i=1}^{C} -p_i \log(p_i) \tag{1}$$

The partial derivative of $h_i$ corresponds to softmax vector $p$ is

$$\frac{\nabla \mathcal{L}_{EM}}{\partial h_i} = p_i(1 - p_i)\log(p_i) + \sum_{j \neq i}^{C} p_i p_j \log(p_j) \tag{2}$$

Then, we can evaluate the $L_1$ gradient norm of sample $x$

$$L_{grad}^{EM}(h) = \sum_{i=1}^{C} \left| p_i(1 - p_i)\log(p_i) + \sum_{j \neq i}^{C} p_i p_j \log(p_j) \right| \tag{3}$$

**Cross-entropy**  Similar to entropy minimization loss, cross-entropy loss is defined

$$\mathcal{L}_{CE} = \sum_{i=1}^{C} -\hat{y}_i log(p_i) \tag{4}$$

Applying the chain rule on cross-entropy, we achieve its partial gradient for each logit $h_i$ as follows

$$\frac{\nabla \mathcal{L}_{CE}}{\partial h_i} = -\hat{y}_i + p_i \tag{5}$$

The $L_1$ gradient norm of cross-entropy loss corresponding to logit $h$ will be

$$L_{grad}^{CE}(h) = 2(1 - p_k) \tag{6}$$

where class $k$ is the pseudo-class of input sample $x$.

## 3.2 STABLE LEARNING MECHANIC: MOVING FROM SOFTMAX SPACE TO LOGIT SPACE

First of all, we discuss the limitation of the original cross-entropy loss from the gradient perspective. Equation 5 reveals that cross-entropy loss pays more attention to hard negative classes (classes with high probability but not the pseudo-class) and does not focus on the pseudo-class when it already achieves high probability. Specifically, the magnitude of the derivative corresponding to class $k$ (where $k$ is not the pseudo-class) is $|p_k - y_k| = p_k$, which indicates that negative classes with high probability will be updated faster. On the other hand, for the pseudo-class (class $k$), its derivative magnitude is $1 - p_k$. Consequently, when $p_k$ approaches 1, the derivative update converges to 0. This updating mechanism, however, is not efficient under pseudo-label learning. When the probability of a pseudo label is close to 1, it means there is a higher chance this predicted label will be the true one. Therefore, a good loss function should still pay attention to these pseudo-classes during the updating, instead of discarding them.

In order to overcome the above limitation and yield a stable training effect, we would like to propose a surrogate loss, which is motivated by cross-entropy. Back to Equation 4, the probability of class $i$: $p_i$ is defined

$$p_i = \frac{\exp(h_i)}{\sum_{j=1}^{C} \exp(h_j)} \tag{7}$$

So we have

$$p_i \approx \exp(h_i) \tag{8}$$

Replacing the Equation 8 in to Equation 4 we acquire a new loss function

$$\mathcal{L}_{sparse-CL} = -\sum_{i}^{C} \hat{y}_i \log(\exp(h_i)) = -\sum_{i}^{C} \hat{y}_i h_i \tag{9}$$

The partial derivative of class $i$ will be

$$\frac{\nabla \mathcal{L}_{sparse-CL}}{\partial h_i} = -\hat{y}_i \tag{10}$$

We name this new loss function sparse cross-logit (sparse-CL). This loss function overcomes the limitation of cross-entropy by forcing the model to update a pseudo-class with a derivative equal to $\hat{y}_i$, instead of assigning it approximately 0 when $\hat{y}_i$ approaches 1, and this helps the model learn efficiently from pseudo labels.

Moreover, the $L_1$ gradient norm of this loss concerning logit $h$ is

$$L_{grad}^{sparse-CL}(h) = \sum_{i}^{C} |\hat{y}_i| = 1 \tag{11}$$

Compared with Equations 3 and 6, we find that the variance of the gradient norm respect to $h$ when learning with sparse-CL is equal to 0 (due to the constant $L_1$ gradient norm). This indicates that this loss will yield a smaller gradient variance during updating and a stable gradient norm in the backward steps. As a result, we can adapt the model with this loss using a high learning rate. In Figure 3, we visualize that our loss function yields a more stable $L_1$ gradient norm compared to entropy minimization and cross-entropy. In the experiment section, we will show that with this simple loss function, we can effectively train a network using a high learning rate without hurting model performance. Additionally, this loss introduces sparse updating, where the gradient updates only along the pseudo-class, while the other classes will have zero gradient updates, resulting in efficient updating.

### 3.3 IMPROVE THE DISCRIMINATIVE ABILITY: K-HARDNESS NEGATIVE LEARNING

Despite its positive effect, the sparse-CL loss has a limitation: during the backward pass, the gradient flows only through the highest classes, while ignoring the others (gradients are equal to zero). This updating may be efficient in terms of computation cost. However, we also expect that this loss can utilize information from both the pseudo-class (which we set as a positive class) and the other classes

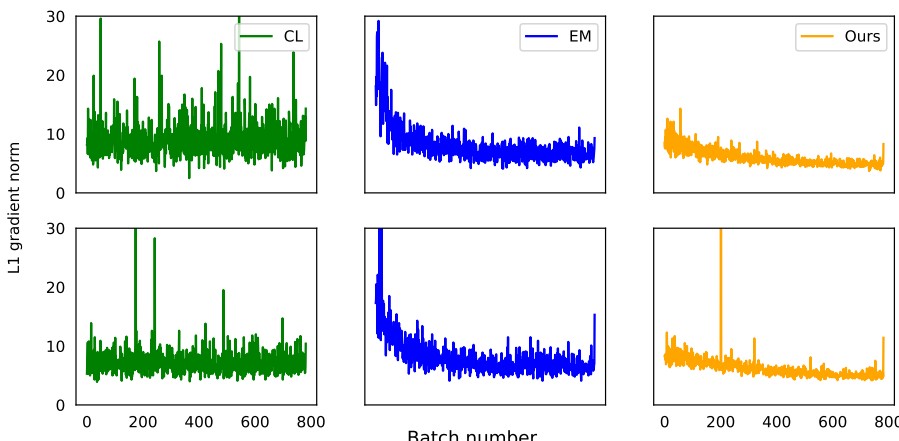

Figure 3: The gradient norm of cross-entropy (left), entropy minimization (middle), and our loss method (right) on two different corruptions: Gaussian noise (first row) and Defocus blur (second row) of ImageNet-C, severity level 5. Our method achieves a more stable gradient norm, while the gradient norm of cross-entropy and entropy minimization fluctuates significantly during the learning process. This stability is the key to the success of our method when learning with a high learning rate.

---

**Algorithm 1** Stable Test-time adaptation for noisy-pseudo labels

**Require:**
1: **Data**: Test sample $X_t = \{x_j\}_{j=1}^M$, class labels: $\{1, 2, \cdots, C\}$.
2: **Model**: Target model $F_t(\cdot)$ with parameters $\theta$, and trainable parameters $\phi$ ($\phi \subseteq \theta$). Besides, we decompose $F_t(\cdot)$ into the feature extractor $f_t(\cdot)$ and the classifier $h_t(\cdot)$.
3: **Hyperparameters**: $k$ is the number of complementary labels considered, and $s$ specifies the number of logits skipped to avoid noise when constructing complementary labels. Here, a complementary label specifies a class that a test sample does not belong to (e.g. the complementary label for a dog image is: cat, elephant,...), learning rate $\eta$ will determine the updating magnitude, and controllable weight $\alpha$ will control the balance between sparse-CL and k-NL losses.

**Ensure:** Prediction labels $\{\hat{y}_j\}_{j=1}^M$ for test samples.
4: Initialize $\theta = \theta_s$          ▷ Initialize model parameters $\theta$ with source model parameters $\theta_s$
5: **for** $x_j \in X_t$ **do**
6:      Calculate logits: $h = h_t(f_t(x_j))$
7:      Sort logits in descending order: $h_{sorted} = Sort(h)$
8:      Assign the pseudo-label to the class with the highest logit: $\hat{y} = One\_hot(h_{sorted}[1])$
9:      Skip the top $s$ logits (from index 2) to avoid noise: $h_{skipped} = h_{sorted}[s + 3 :]$
10:      Select the next $k$ logits in $h_{skipped}$ as negative classes: $h_{negative} = h_{skipped}[1 : k + 1]$
11:      Generate complementary labels: $\overline{y} = Complementary\_labels(h_{negative})$     ▷ The Complementary_labels() function will create a binary vector with length $|C|$, where the index of complementary classes (classes in $h_{negative}$ will be assign value 1, and 0 for the others)
12:      Compute sparse-CL loss: $\mathcal{L}_{sparse-CL} = -\sum_i^C \hat{y}_i h_i$         ▷ Equation 9
13:      Compute k-NL loss: $\mathcal{L}_{k-NL} = \sum_i^C \overline{y_i} h_i$         ▷ Equation 14
14:      Compute final loss: $\mathcal{L}_{final} = \alpha \times \mathcal{L}_{sparse-CL} + (1 - \alpha) \times \mathcal{L}_{k-NL}$    ▷ Equation 17
15:      Compute gradient: $g = \nabla \mathcal{L}_{final}$
16:      Update the model parameters: $\phi \leftarrow \phi - \eta g$
17: **end for**

---

(which we call negative classes) jointly to learn together. On the one hand, learning with a positive class helps the model explore semantic features along that class. On the other hand, learning using negative classes helps the model create clear decision boundaries between samples and their hard negatives. To inject negative class information, we utilize the negative learning approach from (Kim et al., 2019; 2021), where they define negative learning as an auxiliary loss function that can help the overall learning process become more accurate. Their negative loss function is defined as follows

$$\mathcal{L}_{NL+}(f, \overline{y}) = -(1 - p_{\overline{y}}) \sum_{k=1}^{C} \overline{y}_k \log(1 - p_k) \tag{12}$$

where $\overline{y}$ is the negative label, which is defined using $k$ hardest negative sample in the complementary set $\overline{C}$ (which mean $i \in \{1, 2, \cdots, k\}$, $\overline{y}_i = 1$, if $\in \{1, 2, \cdots, k\}$, $\overline{y}_j = 0$, otherwise). And $f$ is the model we want to optimize, its gradient will be

$$\nabla \mathcal{L}_{NL+(i=\overline{y})} = \begin{cases} (1 - p_{\overline{y}})p_i, & if \quad i = \overline{y} \\ -p_{\overline{y}}p_i, & if \quad i \neq \overline{y}_i \end{cases} \tag{13}$$

where $i \in \{1, 2, \cdots, C\}$ is the index of $i$-th classes.

$k$-**hardness negative loss**  Based on the gradient analysis in Equation 13, we can see that the gradient of class $i$ will be proportional to its probability magnitude. This means the model will focus on hard negative samples rather than the easy ones (hardness-aware loss), and this characteristic helps the training model learn better features (Wang & Liu, 2021). However, the above loss function has two limitations: **1.** Unstable gradient norm (its $L_1$ gradient norm will be equal to $\sum_{i \neq \overline{y}}^{C} | - p_{\overline{y}}p_i| + |(1 - p_{\overline{y}})p_{\overline{y}}| = 2p_{\overline{y}} \sum_{i \neq \overline{y}}^{C} p_i$). **2.** For classes that have low probabilities, the target model still updates them with a small amount (in terms of gradient). This trait, on the one hand, slows down the updating process (the gradient needs to go through all classes). On the other hand, we argue that negative information from the hard negative classes is enough, and updating on the other classes can make the model harder to converge.

To tackle these limitations, we propose another loss, which can satisfy the goal of negative loss and overcome its drawbacks

$$\mathcal{L}_{k-NL} = \sum_{i}^{C} \overline{y}_i h_i \tag{14}$$

where $\overline{y}$ is the negative labels, Analyze the gradient of Equation 14 we achieve

$$\frac{\nabla \mathcal{L}_{k-NL}}{h_i} = \begin{cases} 0, & if \quad \overline{y}_i = 0 \\ 1, & if \quad \overline{y}_i = 1 \end{cases} \tag{15}$$

and its $L_1$ gradient norm is

$$L_{grad}^{k-NL}(h) = \sum_{i=1}^{C} \left| \frac{\nabla \mathcal{L}_{k-NL}}{\partial h_i} \right| = k \tag{16}$$

We name this loss $k$-**hardness negative loss** ($k$-NL). It also works in the logit space and yields zero gradient variance during adaptation with respect to the logit $h$ (its $L_1$ gradient norm is equal to $k$, the number of selected negative samples). This negative loss uses multiple complementary labels instead of a single one, as in the original negative loss. Moreover, these complementary labels are selected by first skipping the $s$ highest probabilities in the complementary set (which have been sorted in descending order of probability), and then choosing the next $k$ highest samples (in terms of probability) as the complementary ones, instead of random selection. Discarding the first $s$ nearby classes helps reduce the noisy effect during the learning process of negative loss, which can hurt the model during adaptation. Additionally, the gradient in Equation 15 shows that selecting $k$ complementary labels in this way keeps the hardness-aware characteristic of the original negative loss unchanged. Furthermore, the gradients of the remaining classes in the complementary set are equal to zero, which can reduce the updating cost and ensure the sparse updating characteristic in the sparse-CL loss when combined.

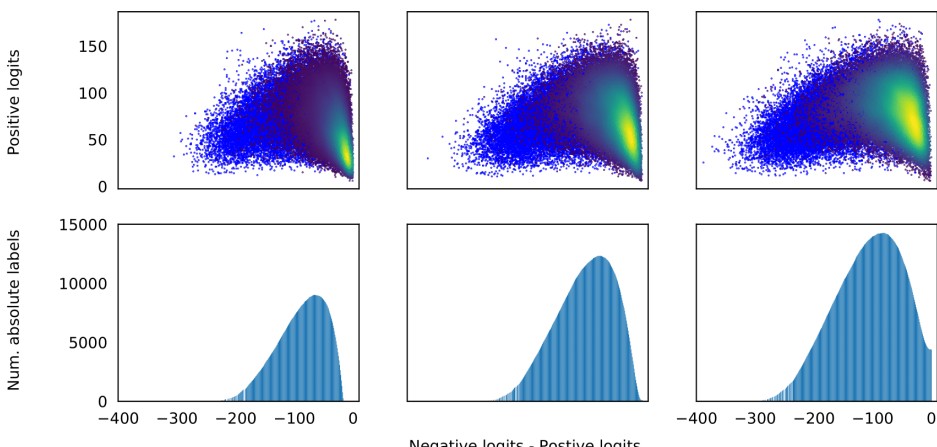

Figure 4: In this figure, we aim to further understand the behavior of predicted labels in the logit space by utilizing positive (magnitude of sparse-CL loss) and negative (magnitude of $k$-NL) logits running on three losses: entropy minimization (left), sparse-CL (middle), and sparse-CL + $k$-NL (right) under the Gaussian noise corruption of ImageNet-C. Generally, the first row visualizes how true pseudo labels (blue) and false (noise) pseudo labels (dark blue) are distributed. For noisy labels, we further plot the label density region, where the y-axis is the magnitude of positive logits, and the x-axis is the difference in magnitude between negative logits ($k$-NL loss) and positive logits (sparse-CL loss). Moreover, the second row further explains this correlation by visualizing the accumulation number of absolute true labels (the absolute true labels along the y-axis at x-coordinate $t$ is the number of true label samples minus false (noisy) label samples, where $t$ indicates that we just consider samples with magnitudes between negative and positive logits smaller or equal to $t$). The second row reveals that applying sparse-CL loss helps recognize the true and noisy pseudo labels more easily based on positive and negative logits, and combining it with $k$-NL enhances this trait. Generally, we can summarize these insights as follows: **1.** In the first row, adapting sparse-CL shifts the true labels toward the left side, while forcing the noisy labels to the right side, which shows that the target model generates clearer decision boundaries between positive and negative classes. **2.** Noisy labels, when applied with sparse-CL, tend to lie in the right-most region (the high-density region), and this phenomenon is increased when adding $k$-NL. **3.** In the second row, the number of absolute true samples reaches approximately 9000, 12000, and 14000 samples when using entropy minimization, sparse-CL, and sparse-CL + $k$-NL loss, respectively. This indicates that we could use the difference between negative and positive logits as a criterion to decide whether the pseudo labels of samples are true labels or noisy ones (similar to how previous work (Niu et al., 2022) used the magnitude of entropy to detect highly reliable samples).

**Final loss**   The final loss will be the combining between sparse-CL and $k$-NL loss, which can be defined as follows

$$\mathcal{L}_{final} = \alpha \times \mathcal{L}_{sparse-CL} + (1 - \alpha) \times \mathcal{L}_{k-NL} \tag{17}$$

where $\alpha$ is the hyperparameter that controls the contribution between sparse-CL and $k$-NL losses.

## 4   EXPERIMENTS

**Set up**   To clarify the robustness of our method, we conducted experiments on the ImageNet-C dataset, focusing on 15 corruption types at the highest corruption level (level 5). Specifically, we replaced the entropy minimization loss in SAR with our loss function. We investigated the performance of the proposed loss from two perspectives: sparse-CL when used alone and combined with $k$-NL in various real-world settings. Due to space limitations, we present the results for three main settings in this section: the normal setting (learning with a batch size of 64), the imbalance setting (following SAR, we created an imbalanced version of ImageNet-C and verified our model's

| Model | Gauss. | Shot | Impul. | Defoc. | Glass | Motion | Zoom | Snow | Frost | Fog | Brit. | Contr. | Elastic | Pixel | JPEG | Avg. |
|---|---|---|---|---|---|---|---|---|---|---|---|---|---|---|---|---|
| MEMO | 21.6 | 17.4 | 20.6 | 37.1 | 29.6 | 40.6 | 34.4 | 25.0 | 34.8 | 55.2 | 65.0 | 54.9 | 37.4 | 55.5 | 57.7 | 39.1 |
| DDA | 41.3 | 41.3 | 40.6 | 24.6 | 27.4 | 30.7 | 26.9 | 18.2 | 27.7 | 34.8 | 50.0 | 32.3 | 42.2 | 52.5 | 52.7 | 36.2 |
| EATA | 35.9 | 4.6 | 36.7 | 45.3 | 47.2 | 49.3 | 47.7 | 56.5 | 55.4 | 62.2 | 72.2 | 21.7 | 56.2 | 64.7 | 63.7 | 49.9 |
| TENT† | 48.5 | 47.7 | 49.0 | 54.9 | 52.7 | 58.7 | 54.5 | 9.6 | 14.8 | 69.7 | 76.4 | 66.4 | 59.5 | 69.8 | 67.1 | 53.3 |
| SAR† | 46.5 | 43.1 | 48.9 | 55.3 | 54.3 | 58.9 | 54.8 | 53.6 | 46.2 | 69.7 | 76.2 | 66.2 | 60.9 | 69.6 | 66.6 | 58.0 |
| SAR* + $\mathcal{L}_{sparse-CL}$ | 51.3 | 51.7 | 52.4 | 55.5 | 55.0 | 60.1 | 57.7 | 64.2 | 62.6 | 70.9 | 76.2 | 66.2 | 64.3 | 70.5 | 68.0 | 61.8 |
| SAR* + $\mathcal{L}_{final}$ | 53.1 | 53.8 | 54.3 | 56.5 | 56.7 | 61.8 | 60.3 | 66.1 | 64.4 | 72.0 | 76.7 | 66.7 | 66.7 | 71.7 | 68.9 | 63.2 |

Table 1: Under the imbalance setting, sparse-CL helps SAR improve by $3.8\%$. This improvement comes from two aspects: first, sparse-CL overcomes the collapse cases in which SAR fails (such as snow and frost). Second, this loss boosts the performance of other corruption types, indicating that it guides the model towards a better solution. Furthermore, combining sparse-CL with $k$-NL demonstrates the effectiveness of negative learning, raising the accuracy improvement from $3.8\%$ to $5.2\%$.

| Model | Gauss. | Shot | Impul. | Defoc. | Glass | Motion | Zoom | Snow | Frost | Fog | Brit. | Contr. | Elastic | Pixel | JPEG | Avg. |
|---|---|---|---|---|---|---|---|---|---|---|---|---|---|---|---|---|
| MEMO | 21.6 | 17.3 | 20.6 | 37.1 | 29.6 | 40.4 | 34.4 | 24.9 | 34.7 | 55.1 | 64.8 | 54.9 | 37.4 | 55.4 | 57.6 | 39.1 |
| DDA | 41.3 | 41.1 | 40.7 | 24.4 | 27.2 | 30.6 | 26.9 | 18.3 | 27.5 | 34.6 | 50.1 | 32.4 | 42.3 | 52.2 | 52.6 | 36.1 |
| EATA | 29.7 | 25.1 | 34.6 | 44.7 | 39.2 | 48.3 | 42.4 | 37.5 | 45.9 | 60.0 | 65.9 | 61.2 | 46.4 | 58.2 | 59.6 | 46.6 |
| TENT† | 49.2 | 48.2 | 49.4 | 55.1 | 53.2 | 59.1 | 54.6 | 11.4 | 12.4 | 69.8 | 76.4 | 66.7 | 60.8 | 70.0 | 67.4 | 53.6 |
| SAR† | 45.4 | 42.4 | 46.0 | 53.7 | 50.4 | 57.7 | 52.8 | 59.0 | 58.0 | 69.0 | 76.0 | 65.8 | 58.5 | 68.8 | 66.3 | 58.0 |
| SAR* + $\mathcal{L}_{sparse-CL}$ | 53.5 | 54.3 | 54.6 | 55.9 | 57.1 | 62.8 | 62.3 | 66.6 | 64.3 | 71.3 | 75.4 | 65.7 | 68.1 | 71.3 | 68.3 | 63.4 |
| SAR* + $\mathcal{L}_{final}$ | 55.0 | 56.1 | 56.2 | 56.9 | 58.4 | 64.3 | 64.4 | 67.8 | 65.4 | 72.3 | 76.1 | 66.2 | 69.6 | 72.3 | 69.1 | 64.7 |

Table 2: Under the small batch size learning setting (batch size equal to 1), our method can achieve an improvement of up to $6.7\%$ compared to using entropy minimization loss. Similar to imbalance learning, this setting is one of the most challenging when working with TTA. Therefore, these promising results highlight our method, which could be used as a baseline loss for TTA.

performance in this setting; [1], with a batch size of $64$), and the small batch size setting, where we investigated our model's performance with a batch size of $1$. Additional experimental results are presented in the Supplementary material.

**Result** We compare our method with recent state-of-the-art methods [2] [3]: MEMO (Zhang et al., 2022a) augments multiple copies of test samples, then minimizes the marginal entropy. DDA (Gao et al., 2022) utilizes a diffusion model to tackle the TTA problem. TENT (Wang et al., 2020) updates the BN layers through entropy minimization. (Niu et al., 2022) further improves TENT by adapting high entropy sample filtering and Fisher regularization to mitigate catastrophic forgetting. Additionally, (Niu et al., 2023) addresses real-world TTA with sharpness-aware loss and model recovery. Generally, the results show that simply adapting sparse-CL can outperform the previous entropy minimization loss by a large margin. Combining it with $k$-NL training further boosts model performance. We summarize the key experimental results of our method here (more detailed results are presented in Tables 1, 2, and 3).:

- Sparse-CL significantly improves SAR performance across all settings. Additionally, it overcomes the collapse phenomenon associated with entropy minimization loss.

---

[1] more details can be found in the Supplementary material.

[2] Model with †: the result is reproduced.

[3] SAR*: we utilize the SAR setting and replace self-entropy minimization loss with our one.

| Model | Gauss. | Shot | Impul. | Defoc. | Glass | Motion | Zoom | Snow | Frost | Fog | Brit. | Contr. | Elastic | Pixel | JPEG | Avg. |
|---|---|---|---|---|---|---|---|---|---|---|---|---|---|---|---|---|
| TENT† | 45.0 | 43.4 | 45.5 | 52.4 | 48.2 | 55.6 | 51.3 | 26.7 | 24.0 | 66.7 | 75.2 | 64.9 | 54.0 | 67.1 | 64.7 | 52.3 |
| SAR† | 45.8 | 44.2 | 47.0 | 53.0 | 49.9 | 55.8 | 51.5 | 57.4 | 56.3 | 66.2 | 74.8 | 64.4 | 55.0 | 66.9 | 64.4 | 56.8 |
| SAR* + $\mathcal{L}_{sparse-CL}$ | 53.3 | 54.0 | 54.4 | 56.1 | 57.2 | 62.4 | 61.1 | 65.9 | 64.1 | 71.4 | 75.8 | 65.8 | 67.4 | 71.0 | 68.3 | 63.2 |
| SAR* + $\mathcal{L}_{final}$ | 56.0 | 57.0 | 57.1 | 58.6 | 59.7 | 63.7 | 63.6 | 67.3 | 65.2 | 72.1 | 76.1 | 66.6 | 69.1 | 72.0 | 69.1 | 64.9 |

Table 3: Learning with the normal setting yields the biggest improvement among all settings. Our loss helps SAR increase from $56.6\%$ to $63.8\%$ (a $6.4\%$ increase) and climb to $64.8\%$ (an $8.1\%$ improvement) when jointly learned with $k$-NL. Combining these results with those from the challenging settings (imbalance, small batch size), we can conclude that our method is suitable for various real-world data settings, as indicated by the stable and significant improvements when adapted to them.

- Combining Sparse-CL with $k$-NL further enhances performance across all learning settings, indicating the positive impact of $k$-NL. It could serve as a valuable auxiliary task to improve model learning.

- Figure 4 illustrates why our loss function performs well under Sparse-CL and $k$-NL learning. The distinction between true (positive) and noisy (negative) labels becomes clearer when examining the difference in logit magnitudes between positive and negative classes. This evidence supports the hypothesis that learning with our loss function can prevent misclassification of nearby classes, resulting in a clear decision boundary across classes.

## 5 CONCLUSION

In this work, we propose a stable learning strategy to mitigate the noise problems of pseudo labels under TTA learning. Our study first analyzes the main reasons for the failure of current learning methods using pseudo labels. We then highlight that gradient variance is a key factor in stable learning and introduce a new loss function named Sparse-CL, inspired by cross-entropy loss, which ensures stable updates. Our training strategy updates the model with a high learning rate, hypothesizing that this rapid updating helps the model adapt well to the target domain and effectively combat both memorization and confirmation bias problems. Additionally, we enhance cross-logit learning by incorporating a negative loss named $k$-NL, which leverages complementary labels and helps the model focus on k-hardness negative samples, thereby improving overall model performance.

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
