# OpenReview forum: "Update larger, train faster: Stable Test-time adaptation utilizing noisy-pseudo labels"
_ICLR.cc/2025/Conference — Submitted to ICLR 2025_

### Official Review · Reviewer_j81H · 2024-10-17

**Soundness:** 2
**Presentation:** 1
**Contribution:** 2
**Rating:** 3
**Confidence:** 5

**Summary:**

The paper proposes a new approach for test-time adaptation using pseudo-labels to address challenges such as memorization effect and confirmation bias. The method consists of a sparse cross-logit for learning top-1 label prediction, with k-hardness negative learning to penalize other labels. The experiment on ImageNet-C demonstrates the improvement over baseline methods across various settings.

**Strengths:**

- The proposed method shows a substantial performance gain on ImageNet-C.
- The approach is relatively simple to implement and integrate with existing TTA methods.

**Weaknesses:**

**Lack of justification for the method.**
- It is not straightforward how the proposed methodology tackles the memorization effect and confirmation bias. Also, the memorization effect has been discussed in the centralized training setup, necessitating further justification for the test-time adaptation setting.
- The transition from Equation 7 to 8 is vague. The paper simply ignores the denominator in the softmax calculation without any justification.
- Comparing Equation 11 with Equation 6, it is unclear how the gradient of sparse-CL would lead to a higher learning rate of 100x~5000x.
- Equation 6 is already bound in [0, 2], where confident samples would have softmax probability $p_i$ near 1. This comparison suggests original cross-entropy can use a higher learning rate, which contradicts the paper.


**The generalizability of the method is questionable.**
- The paper does not justify selecting learning rates, multiple hyperparameters for k-NL loss ($s, k$), and balancing hyperparameter $\alpha$.
- Applying the method only on SAR (and TENT) is vague. Especially the impact of sample filtering and sharpness-aware minimization in SAR is not properly discussed.
- Experimental results demonstrate good performance improvement on ImageNet-C but a highly marginal improvement on CIFAR100-C, limiting the generalizability among the datasets.
- The paper lacks recent test-time adaptation baselines (e.g., RoTTA [a], SoTTA [b], DeYO [c]).


**Poor writing quality.**
- Figure 2: The caption and legend mismatches.
- Figure 3: The gradient norm differs from the L1-norm explained in the main text. How did the authors calculate the gradient norm? Also, the displayed contents do not show a high difference between entropy minimization and the paper's method.
- Figure 4: Each sub-figure is poorly displayed, and it is hard to grab the message from the figure. The current figure is insufficient to support the authors' three findings.
- Algorithm 1: Need a better display with concise explanations.
- Appendix B.1: "Sparse updating supports faster training (fewer parameters need to be updated)" needs further justification and contradicts Table 7.
- Appendix: Please change the numbering of Figures to avoid overlapping with the main manuscript.


**The paper does not claim reproducibility.**
- The paper does not contain the code or explicitly claims about reproducibility.

---

References
- [a] Yuan, Longhui, Binhui Xie, and Shuang Li. "Robust test-time adaptation in dynamic scenarios." Proceedings of the IEEE/CVF Conference on Computer Vision and Pattern Recognition. 2023.
- [b] Gong, Taesik, et al. "SoTTA: Robust Test-Time Adaptation on Noisy Data Streams." Advances in Neural Information Processing Systems. 2024.
- [c] Lee, Jonghyun, et al. "Entropy is not Enough for Test-Time Adaptation: From the Perspective of Disentangled Factors." The Twelfth International Conference on Learning Representations. 2024.

**Questions:**

- Why should we skip $s$ classes for k-NL? Any theoretical or empirical observations of the intuition beyond Figure 2 in the Appendix?
- How should we select $s$ and $k$? How does $s, k$ affect the overall stability?

---

### Official Review · Reviewer_BfjN · 2024-10-31

**Soundness:** 2
**Presentation:** 1
**Contribution:** 2
**Rating:** 3
**Confidence:** 3

**Summary:**

This paper identifies the issues with using pseudo-labels in Test-Time Adaptation (TTA), specifically the "slow" adaptation to the target domain. To address this, the authors propose Sparse-CL, which enables larger learning steps for quicker adaptation. Experiments validate the effectiveness of the proposed method.

**Strengths:**

This paper discusses the challenges of using pseudo-labels in Test-Time Adaptation (TTA) and provides a detailed analysis of the limitations of existing methods.

The experiments demonstrate the high performance of the proposed approach.

**Weaknesses:**

1. The writing is poor, e.g.
    - Line 235. “So we have”. How to get Eq. (8) from Eq. (7). ?
    - Line 334, “$\in \{1, 2, \cdots, k\}, \bar y_j =0$”, what does that means? Also, what is $k$ and $j$ here?
    - Eq (13), "if $ i \neq \bar y_i$". What does that mean?
    - Eq (15). "$\frac{ \nabla \mathcal{L}_{k-NL}}{h_i}$".
    - The captions of all the tables are too long, these contents should be included in the main text.
2. The author states that they can adapt the model with their loss using a high learning rate. However, there is no analysis of the learning rate in the experiment. The only related results are in the introduction with limited implement details.
3. This paper lacks of ablation study of the hyper-parameters (i.e., $s$, $k$, $a$).

**Questions:**

1. Can you provide some evidence to validate the effectiveness of “skipping the $s$ highest probabilities”?

1. Table 1. “first, sparse-CL overcomes the collapse cases in which SAR fails (such as snow and frost). Second, this loss boosts the performance of other corruption types, indicating that it guides the model towards a better solution.” Any evidence?

---

### Official Review · Reviewer_ARHE · 2024-11-03

**Soundness:** 3
**Presentation:** 2
**Contribution:** 2
**Rating:** 5
**Confidence:** 5

**Summary:**

This paper revisits the problem of error accumulation due to domain changes in the CTTA task from the perspective of the speed of model adaptation, which is a novel perspective, and the authors further propose Sparse-CL for fast adaptation, which in turn proposes K-NL loss to compensate for Sparse-CL's dependence on pseudo-labeling.

**Strengths:**

Considering the error accumulation problem of CTTA from the perspective of adaptation speed is a very novel perspective, providing a new entry direction for solving the CTTA problem, while Sparse-CL can be easily integrated into various existing frameworks.

**Weaknesses:**

The text does not provide a more detailed explanation of the starting point of the method, while the theoretical proof of why sparse-cl works is incomprehensible.
limitations：
1. The title of the paper emphasizes larger and faster, does this mean that the model will adapt faster? But all the test data in the TTA scenario need to be adapted, so why would faster convergence be achieved?
2. Why does faster convergence alleviate the error accumulation problem of CTTA? The authors do not seem to provide enough explanation for this crucial starting point.
3. As for the design of the sparse-cl loss function, it has been emphasized in the paper that the nature of its gradient paradigm can provide a more robust adaptation, why can it achieve a more robust effect? Is there any theoretical basis for this? The authors need to provide further explanation.
4. Since the authors have emphasized the advantage of its convergence speed, whether too fast convergence will bring overfitting problem, which needs to be discussed and analyzed.
5. In K-NL, it is necessary to skip the 2nd to s+1st intermediate samples and then take k samples, why is it designed this way? Why not just take k samples?
6. In the experimental section, you need to provide further split experiments to verify the validity of your method. Also, it is important to analyze the hyperparameters, because I can't tell if your method relies heavily on the choice of s and k.

**Questions:**

See the weakness.

---

### Official Review · Reviewer_Zkux · 2024-11-04

**Soundness:** 3
**Presentation:** 2
**Contribution:** 3
**Rating:** 5
**Confidence:** 5

**Summary:**

This paper aims to address the challenges of the memorization effect and confirmation bias in pseudo-label learning for test-time adaptation due to noise issues. The authors compare cross-entropy, entropy minimization, and losses computed in the logits space, and find that the key factor leading to unstable learning is gradient variance. Based on this, the authors propose a new loss function, Sparse-CL. Additionally, the paper introduces a negative loss named k-NL as a contrastive loss to generate a clear decision boundary. Experimental results show that the proposed loss functions achieve superior performance in both imbalance setting and small batch size learning setting. The sparse updating enables the proposed method to learn quickly under large learning rate conditions.

**Strengths:**

This paper provides a novel approach to modeling the test-time adaptation problem as a noisy pseudo-label learning task. It offers an interesting insight by analyzing the shortcomings of cross-entropy and entropy minimization from the perspective of gradient variance. The proposed loss function is simple and effective, and it can be easily scaled to other TTA methods.

**Weaknesses:**

1. The proposed sparse-CL and k-NL mask the loss for some classes, controlled by the hyperparameters $s$ and $k$. Therefore, the choice of $s$ and $k$ is crucial for the effectiveness of the proposed method. However, in Supplementary Material A.3, it is only stated that $s$ and $k$ are chosen as $5$ based on cross-validation. This empirical selection reduces the generalizability of the proposed loss function.

2. Additionally, the paper lacks an explanation and ablation study for the value of the hyperparameter $\alpha$ in Equation 17.

3. The derivation from Equation 7 to Equation 8 appears to be problematic. Assuming the denominator in Equation 7 is approximately equal to 1 to derive Equation 8 is incorrect.

4. Typo: line 334 in the main text, “if ∈ {1, 2, · · · , k}” seems to be redundant.

**Questions:**

See Weakness. If the authors can address my concerns, I will raise my score.

---

### Meta-Review · Area_Chair_d7Mw · 2024-12-19

**Metareview:**

This paper seeks to tackle the challenges of the memorization effect and confirmation bias in pseudo-label learning for test-time adaptation caused by noise-related issues. However, the reviewers raised numerous questions regarding the experimental section, particularly about hyper-parameters and ablation studies, and expressed concerns about the clarity of the writing and overall presentation. As a result, I would like to suggest a rejection to this manuscript.

**Additional Comments On Reviewer Discussion:**

The authors haven't provided any responses to the concerns raised by the reviewers.

---

### Decision · Program_Chairs · 2025-01-22

Reject